

# Spatial and temporal variability of turbulence dissipation rate in complex terrain

Nicola Bodini[1], Julie K. Lundquist[1,2], Raghavendra Krishnamurthy[3], Mikhail Pekour[4], and Larry K. Berg[4]

[1]Department of Atmospheric and Oceanic Sciences, University of Colorado Boulder, Boulder, Colorado, USA
[2]National Renewable Energy Laboratory, Golden, Colorado, USA
[3]University of Notre-Dame, Notre-Dame, Indiana, USA
[4]Pacific Northwest National Laboratory, Richland, Washington, USA

**Correspondence:** Nicola Bodini (nicola.bodini@colorado.edu)

**Abstract.** To improve the parametrizations of turbulence dissipation rate ($\epsilon$) in numerical weather prediction models, the temporal and spatial variability of $\epsilon$ must be assessed. In this study, we explore influences on the variability of $\epsilon$ at various scales in the Columbia River Gorge during the WFIP2 field experiment between 2015 and 2017. We calculate $\epsilon$ from five sonic anemometers all deployed in a $\sim 4\,\mathrm{km}^2$ area; and from two scanning Doppler lidars and four profiling Doppler lidars, whose locations span a $\sim 300\,\mathrm{km}$ wide region. We retrieve $\epsilon$ from the sonic anemometers using the second-order structure function method, from the scanning lidars with the azimuth structure function approach, and from the profiling lidars with a novel technique using the variance of the line-of-sight velocity. Turbulence dissipation rate shows large spatial variability, even at the microscale, especially during nighttime stable conditions. Orographic features have a strong impact on the variability of $\epsilon$, with the correlation between $\epsilon$ at different stations being highly influenced by terrain. $\epsilon$ shows larger values in sites located downwind of complex orographic structures or in wind farm wakes. A clear diurnal cycle in $\epsilon$ is found, with daytime convective conditions determining values over an order of magnitude higher than nighttime stable conditions. $\epsilon$ also shows a distinct seasonal cycle, with differences greater than an order of magnitude between average $\epsilon$ values in summer and winter.

## 1 Introduction

Numerical weather prediction models currently assume that the generation of turbulence within a grid cell is equal to the dissipation of turbulence $\epsilon$ within the same grid cell. While this assumption, which is appropriate for homogeneous and stationary flow (Albertson et al., 1997), can generally be considered valid when adopting a coarse grid (Lundquist and Chan, 2007; Mirocha et al., 2010), it breaks down when using models with finer horizontal resolution (Nakanishi and Niino, 2006; Krishnamurthy et al., 2011; Hong and Dudhia, 2012), as turbulence can be advected to a different grid cell before being dissipated. However, the scales at which the assumption of local equilibrium is not valid anymore are currently not well understood, as well as how different atmospheric and topographic conditions can impact the development and decay of turbulent structures.

A more accurate representation of turbulence is crucially needed as it represents the fundamental process to transfer heat, momentum and moisture in the atmospheric boundary layer (Garratt, 1994). Moreover, turbulence controls a wide range of





processes with a direct effect on our socio-economic activities: turbulence has impacts on forest fire development and propagation (Coen et al., 2013), it affects air traffic control with its influence on aviation meteorology and the dissipation of aircraft vortices (Gerz et al., 2005; Thobois et al., 2015), it determines the characteristics and impacts of pollutant dispersion (Huang et al., 2013), and it affects wind energy production and the lifetime of wind turbines themselves (Kelley et al., 2006). Moreover,

turbulence dissipation rate has been shown to have a primary role in the formation of frontal structures (Piper and Lundquist, 2004), the evolution of cyclones (Bister and Emanuel, 1998), and the development of flows in urban areas and other canopies (Baik and Kim, 1999; Lundquist and Chan, 2007). The precision of wind energy forecasting is also highly impacted by the accuracy of the representation of turbulence dissipation rate. A recent sensitivity study (Yang et al., 2017; Berg et al., 2019) showed that up to 50% of the variance in the turbine-height wind speed predicted by the Weather Research and Forecast-

ing model (Skamarock et al., 2005) in complex terrain only depends on the accuracy of the parametrization of turbulence dissipation rate.

Various techniques have been developed to calculate $\epsilon$ from different instruments. In general, all the proposed methods are based on the turbulence theory by Kolmogorov (1941), which represents the decay of turbulence eddies as an energy cascade in the inertial subrange, until the length scales are small enough for the turbulence kinetic energy to be dissipated by molecular

diffusion in the viscous subrange. Turbulence dissipation can be calculated from sonic anemometers on meteorological towers (Champagne et al., 1977; Oncley et al., 1996), and super high-frequency hot-wire anemometers suspended on tethered lifting systems (Frehlich et al., 2006; Lundquist and Bariteau, 2015), flown on aircrafts (Fairall et al., 1980) or UAVs (Lawrence and Balsley, 2013). Remote sensing instruments can provide additional insights into our understanding of turbulence dissipation by combining measurements at greater altitudes with their ease of deployment in complex terrain, despite their potential

drawbacks of a limited temporal frequency and their inherent volume averaging (Frehlich and Cornman, 2002; Wang et al., 2016). Wind profiling radars (Shaw and LeMone, 2003; McCaffrey et al., 2017a), profiling lidars, and scanning lidars have all been successfully used to obtain turbulence measurements. For lidars, different approaches have been developed to retrieve $\epsilon$: width of the Doppler spectrum (Smalikho, 1995; Banakh et al., 1995), line-of-sight velocity spectrum (Drobinski et al., 2000; O'Connor et al., 2010; Bodini et al., 2018), structure function (Frehlich, 1994; Banakh et al., 1996; Banakh and Smalikho, 1997;

Smalikho et al., 2005; Frehlich et al., 2006; Wulfmeyer et al., 2016; Smalikho and Banakh, 2017) and range-gate filtering with a subgrid-scale parametrization scheme (Krishnamurthy et al., 2010).

Here, we retrieve turbulence dissipation rate from eleven instruments in a complex terrain region, thus building one of the widest observational assessment of $\epsilon$ to date. We explore how topography triggers the variability of $\epsilon$ at various temporal and spatial scales. We describe the WFIP2 field campaign in Section 2, and we define the characteristics of the sonic anemometers

and wind profiling and scanning lidars that we use to estimate $\epsilon$. We also describe the methods used to retrieve $\epsilon$ from the different instruments, and we further refine and extend a novel approach to derive $\epsilon$ from wind profiling lidars. In Section 3 we present the spatial variability of $\epsilon$ at both the microscale and mesoscale by comparing the estimates from multiple instruments in different locations, with a particular attention to the impact that topography has on the spatial evolution of $\epsilon$. In doing so, we also assess the climatology of turbulence dissipation in terms of both diurnal and seasonal cycles. Section 4 summarizes our





results, and suggests future work to further improve our understanding and representation of turbulence dissipation rate in the boundary layer.

## 2 Data and Methods

### 2.1 The WFIP2 field campaign

5 The Second Wind Forecast Improvement Project (WFIP2) (Shaw et al., 2019), which involved a field campaign (Wilczak et al., 2019) in the U.S. Pacific Northwest between October 2015 and March 2017, was designed to improve numerical weather prediction model forecasts in complex terrain for wind energy applications. A large number of instruments was deployed in the Columbia River Gorge and Basin, in a region over $500\,\mathrm{km}$ wide. In this study, we focus on the evaluation of turbulence dissipation rate from instruments which span an approximately $300\mathrm{km}$ wide area. Two profiling lidars were located at the western and eastern edges of this region, at Troutdale and Vansycle Ridge, respectively, with an additional scanning lidar located in Boardman (Figure 1, panel a). A region with a high-density of instruments (HD Site in Figure 1, panel a), approximately $\sim 20\,\mathrm{km}$ wide, was located in the vicinity of the town of Wasco, from which we will analyze data from two wind profiling lidars and one scanning lidar (Figure 1, panel b) and the sonic anemometers on five meteorological towers (Figure 1, panel c).

Multiple sonic anemometers were located on several meteorological towers at the Physics Site (Wilczak et al., 2019), which 15 represented the finest array of instruments at WFIP2, aimed at having multiple measurements in an area similar in size to a grid cell of a high-resolution numerical weather prediction model. The site, covered by crop fields, is characterized by a moderately complex topography, with terrain elevation spanning from $405\,\mathrm{m}$ to $459\,\mathrm{m}$ ASL (the elevation of the locations of the meteorological towers used in this study are reported in Table 1). Extensive arrays of wind turbines are located on the northern side of the Columbia River and on the south-western part of the studied region, as well as to the east of the Phyiscis 20 Site. The sonic anemometers used in this project provide $20\,\mathrm{Hz}$ measurements of the three components of the wind and virtual temperature at $10\mathrm{m}$ AGL, and they were operational from late-March/early-April 2016 to late-April/mid-May 2017. To account for tower wake effects, data were excluded when the wind direction was within $\pm30°$ from the orientation of the tower boom (McCaffrey et al., 2017b). Less than $10\%$ of the data were excluded due to tower wake contamination.

Data from the sonic anemometers were used to assess atmospheric stability, calculated in terms of the Obukhov length $L$:

$$25 \quad L = -\frac{\overline{\theta_v} \cdot u_*^3}{k \cdot g \cdot \overline{w'\theta_v'}} \tag{1}$$

where $\theta_v$ is the virtual potential temperature ($K$), calculated from the virtual temperature measured by the sonic anemometers; $u_* = (\overline{u'w'}^2 + \overline{v'w'}^2)^{1/4}$ is the friction velocity ($\mathrm{m\,s^{-1}}$); $k = 0.4$ is the von Kármán constant; $g = 9.81\,\mathrm{m\,s^{-2}}$ is the acceleration due to gravity; and $\overline{w'\theta_v'}$ is the kinematic sensible heat flux ($\mathrm{m\,K\,s^{-1}}$). An averaging period of 30 minutes (De Franceschi and Zardi, 2003; Babić et al., 2012) has been used to apply the Reynolds decomposition and determine the fluxes. Based 30 on the value of the Obukhov length, we classify neutral conditions for $L \le -500\,\mathrm{m}$ and $L > 500\,\mathrm{m}$; unstable conditions for $-500\,\mathrm{m} < L \le 0\,\mathrm{m}$; and stable conditions for $0\,\mathrm{m} < L \le 500\,\mathrm{m}$ (Muñoz-Esparza et al., 2012). Neutral conditions were infrequently recorded (less than $10\%$ of the times).



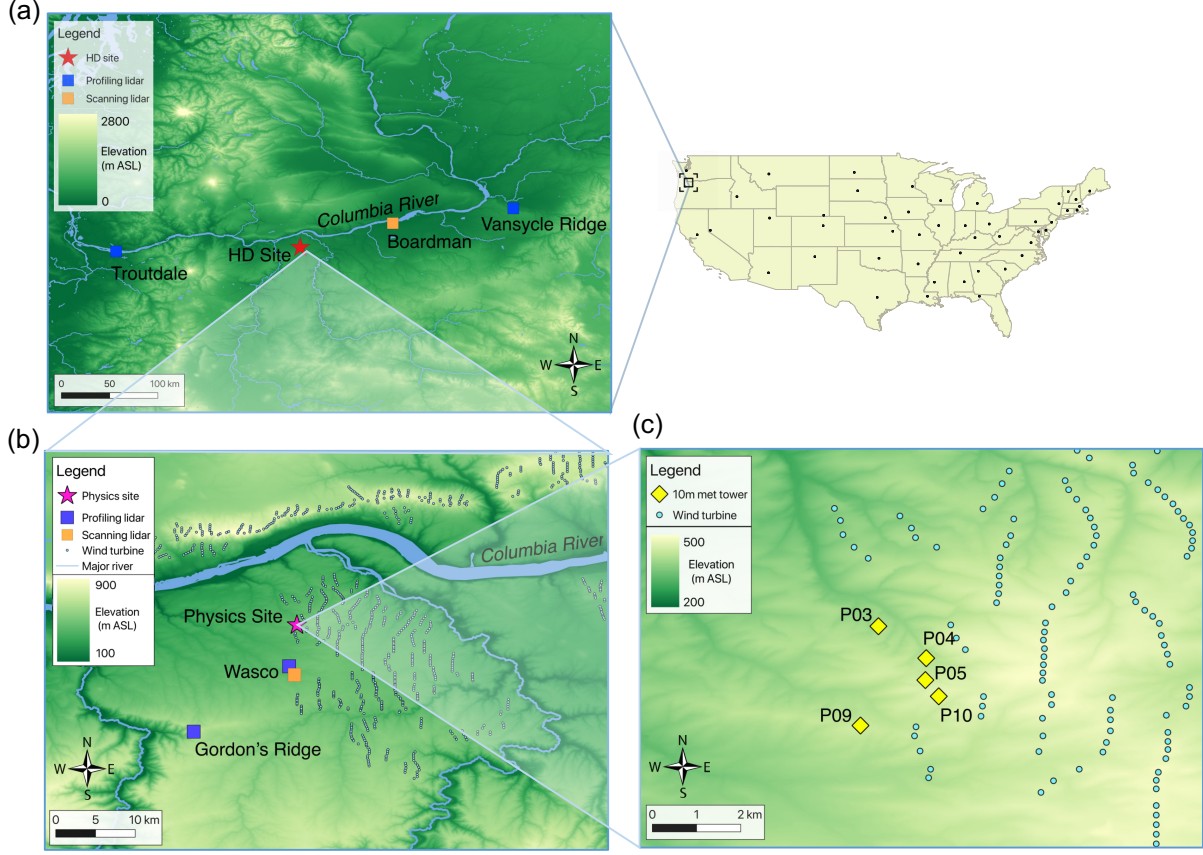

**Figure 1.** Map of the relevant instruments during the WFIP2 field campaign. The locations of the profiling lidars, scanning lidars and meteorological towers used in this analysis are shown.

A WINDCUBE version 1 (v1) was located in Troutdale (12 m ASL), about 20 km east of Portland, in a relatively flat region at the Portland-Troutdale Airport at the western edge of the Columbia River gorge. The area is semi-urban, with some trees. This type of lidar (Aitken et al., 2012; Rhodes and Lundquist, 2013) measures line-of-sight velocity along 4 cardinal directions with a nominal zenith angle of 28° and a temporal resolution of about 1 Hz along each beam direction. The measurements are

5    taken every 20 m, from 40 to 220 m AGL. The main technical specifications of the instrument are shown in Table 2.

A second WINDCUBE v1 and a WINDCUBE 200S scanning Doppler lidar were located at the Wasco Airport (456 m ASL), in an area covered by short grass. The nearby region is characterized by moderately complex topography, in the vicinity of the Columbia River. The WINDCUBE 200S performed a variety of Planned Position Indicator (PPI), Range-Height Indicator (RHI) and vertical stare scans within 15 minutes. Details of the scan patterns can be found in Choukulkar (2018). For this

10    instrument we retrieve $\epsilon$ up to 300 m AGL.

A WINDCUBE version 2 (v2) was deployed on a low-grass surface on the top of Gordon's Ridge (728 m ASL). A second v2 was deployed at Vansycle Ridge (542 m ASL), in a site with grazed grass (Yang et al., 2013) about 20 km east of the





**Table 1.** Elevation and period of data collection of the five 10-m sonic anemometers at the Physics Site, the four profiling lidars and the two scanning lidars considered in this study, whose locations are shown in Figure 1.

| Instrument - site name | Elevation (m ASL) | Data usage period |
|---|---|---|
| Metek sonic anemometer - tower P03 | 405 m | 29 March 2016 - 15 May 2017 |
| Gill sonic anemometer - tower P04 | 426 m | 1 April 2016 - 26 April 2017 |
| Gill sonic anemometer - tower P05 | 449 m | 1 April 2016 - 26 April 2017 |
| Metek sonic anemometer - tower P09 | 438 m | 29 March 2016 - 13 May 2017 |
| Gill sonic anemometer - tower P10 | 459 m | 1 April 2016 - 26 April 2017 |
| WINDCUBE v1 - Troutdale | 12 m | 20 April 2016 - 11 November 2016 |
| WINDCUBE v1 - Wasco | 456 m | 23 February 2016 - 11 November 2016 |
| WINDCUBE v2 - Gordon's Ridge | 728 m | 17 November 2015 - 15 March 2017 |
| WINDCUBE v2 - Vansycle Ridge | 542 m | 10 March 2016 - 17 April 2017 |
| WINDCUBE 200S - Wasco | 456 m | 23 March 2016 - 21 March 2017 |
| Halo Streamline - Boardman | 112 m | 20 April 2016 - 31 August 2016 |

Wallula Gap, where the Columbia River turns north, thus modifying the main topographic direction of the Gorge. Compared to the WINDCUBE version 1, the v2 performs an additional line-of-sight velocity measurement along the vertical, and $\sim 4\,\mathrm{s}$ are required for the beam to complete the five-point scan strategy. The vertical resolution was of $20\,\mathrm{m}$, from 40 to $260\,\mathrm{m}$ AGL ($260\,\mathrm{m}$ AGL for the v2 at Vansycle Ridge). Bodini et al. (2018) compared turbulence dissipation retrievals from co-located WINDCUBE v1s and a v2 and found a good agreement between the different instruments. Table 2 illustrates the major technical parameters of this lidar.

Finally, a Halo Streamline scanning Doppler lidar was deployed near a regional airport surrounded by farmland at Boardman (112m ASL). The long range fiber-optic based scanning Doppler lidar provides 3D scanning capabilities, and performed a wide range of scans covering the atmospheric boundary layer over a period of 15 minutes (Otarola, 2017). In this analysis, only the $5°$ elevation angle scans with a scan rate of $1\,°\mathrm{s}^{-1}$ were used to calculate turbulence dissipation rate up to $120\,\mathrm{m}$ AGL. The other scans within the 15-minute time period were not usable for turbulence calculations due to either fast scan rates or low data availability.

For all the instruments, precipitation periods were excluded from the analysis, based on measurements at two surface meteorological stations at the Wasco airport and Troutdale (for the profiling lidar at that location).

## 2.2 Turbulence dissipation rate from sonic anemometer

We estimate turbulence dissipation rate from the sonic anemometers using the second-order structure function method, which has been demonstrated (Muñoz-Esparza et al., 2018) to provide $\epsilon$ retrievals with a lower error compared to the commonly used inertial-subrange energy spectrum method. The second-order structure function $D_U$ of the horizontal velocity $U$ at the position





**Table 2.** Main technical specifications of the lidars used in this study.

|  | WINDCUBE v1 | WINDCUBE v2 | WINDCUBE 200S | Halo Streamline |
|---|---|---|---|---|
| Wavelength | $1.54\,\mu m$ | $1.54\,\mu m$ | $1.54\,\mu m$ | $1.54\,\mu m$ |
| Receiver bandwidth | $\pm55\,MHz$ | $\pm57.5\,MHz$ | $\pm57.5\,MHz$ | $\pm25\,MHz$ |
| Nyquist velocity ($B$) | $\pm42.3\,m\,s^{-1}$ | $\pm44\,m\,s^{-1}$ | $\pm44\,m\,s^{-1}$ | $\pm19.4\,m\,s^{-1}$ |
| Signal spectral width ($\Delta\nu$) | $3.39\,m\,s^{-1}$ | $2.65\,m\,s^{-1}$ | $1.95\,m\,s^{-1}$ | $1.5\,m\,s^{-1}$ |
| Pulses averaged ($n$) | 10000 | 20000 | 20000 | 10000 |
| Points per range gate ($M$) | 25 | 32 | 64 | 128 |
| Vertical resolution | $20\,m$ | $20\,m$ | $20\,m$ | $20\,m$ |
| Minimum range gate | $40\,m$ | $40\,m$ | $100\,m$ | $60\,m$ |
| Number of range gates | 10 | $9-12$ | 200 | 200 |
| Pulse width | $200\,ns$ | $175\,ns$ | $200\,ns$ | $150\,ns$ |
| Time resolution | $\sim1\,Hz$ | $\sim1\,Hz$ | $1\,Hz$ | $1\,Hz$ |

$x$ is defined as a function of the spatial separation $r$ as $D_U(r) \equiv\, <[U(x+r)-U(x)]^2>$, where $<\cdot>$ denotes an ensemble average. Within the inertial subrange, Kolmogorov's model (Kolmogorov, 1941) relates the second-order structure function with the turbulence dissipation rate $\epsilon$:

$$D_U(r) = \frac{1}{a}\epsilon^{2/3}r^{2/3} \qquad (2)$$

5    where $a$ is the Kolmogorov constant, which we set equal to 0.52 (Paquin and Pond, 1971; Sreenivasan, 1995). By invoking Taylor's frozen turbulence hypothesis (Taylor, 1935), the spatial separation $r$ can be written as temporal separation $\tau$, so that $\epsilon$ can be calculated as:

$$\epsilon = \frac{1}{U\tau}\left[aD_U(\tau)\right]^{3/2} \qquad (3)$$

We calculate $\epsilon$ every $30s$ by fitting the Kolmogorov's theoretical model to the structure function calculated from the sonic
10    anemometer data using a temporal separation between $\tau=0.1\,\mathrm{s}$ and $\tau=2\,\mathrm{s}$. From data inspection, measurements in the chosen time separation interval lie well within the inertial subrange, and therefore they fulfill the hypothesis of Kolmogorov's theory. Moreover, the high temporal resolution of the sonic anemometer suggests an adequate number of data points in this interval to obtain a robust estimate of the structure function.

## 2.3  Turbulence dissipation rate from wind profiling lidar

15    Measurements from wind Doppler lidars can extend our understanding of the variability of turbulence dissipation rate thanks to the their relatively easy deployment even in prohibitive terrain conditions. Moreover, lidars can often provide measurements at higher altitudes compared to most meteorological towers, possibly out of the surface layer.





We follow the approach introduced by O'Connor et al. (2010) and refined by Bodini et al. (2018) to estimate $\epsilon$ from the variance of the line-of-sight velocity measured by the lidars. Assuming locally homogeneous and isotropic turbulence, the one-dimensional spectrum $S$ within the inertial subrange can be written as a function of the wave number $k$ as

$$S(k) = a\epsilon^{2/3}k^{-5/3} \tag{4}$$

where $a = 0.52$ is the one-dimensional Kolmogorov constant. By integrating (4) over the wavenumber space within the inertial subrange, the following expression can be found:

$$\sigma_v^2 = \int\limits_k^{k_1} S(k)dk = -\frac{3}{2}a\epsilon^{2/3}\left(k_1^{-2/3} - k^{-2/3}\right)$$
$$= \frac{3a}{2}\left(\frac{\epsilon}{2\pi}\right)^{2/3}\left(L_N^{2/3} - L_1^{2/3}\right) \tag{5}$$

where $\sigma_v^2$ is the variance of the detrended line-of-sight velocity, and $L_1$ and $L_N$ are the length scales which can be used instead
of the wavenumbers by invoking Taylor's frozen turbulence hypothesis (Taylor, 1935). For a single sample, $L_1$ can be defined as

$$L_1 = Ut + 2z\sin\left(\frac{\theta}{2}\right) \tag{6}$$

where $U$ is the horizontal wind speed, $t$ is the dwell time, $\theta$ is the half-angle divergence of the lidar beam, and $z$ the height AGL. The second term in (6) can typically be neglected as Doppler lidars generally have $\theta < 0.1\,\mathrm{mrad}$. For multiple samples,
$L_N = NL_1$, where $N$ is the number of samples used in the calculation.

The method relies on the fundamental assumption that the samples used in the calculation lie within the inertial subrange of turbulence. If longer samples are used, therefore including contributions from the outer scales, $\epsilon$ will be severely underestimated (Bodini et al., 2018). On the other hand, short samples will undermine the representativeness of the estimation of the turbulence contribution to variance (Lenschow et al., 1994), and a higher relative effect of instrumental noise (Lenschow et al., 2000) will
also increase the error. Therefore, the choice of the sampling size $N$ represents a crucial step to obtain accurate estimates of turbulent quantities, especially in stable conditions (Pichugina et al., 2008).

As shown in Bodini et al. (2018), the appropriate time scales for the lidar retrievals can be determined in different ways. When co-located sonic anemometers are available, the optimal values for $N$ can be found by tuning the lidar method with the $\epsilon$ values derived from the sonic data. Another possibility is the use of spectral models (Kaimal et al., 1972; Panofsky, 1978;
Olesen et al., 1984; Kristensen et al., 1989) to fit the experimental spectra from the lidar measurements and determine the extension of the inertial subrange from the fit (Tonttila et al., 2015).

In the WFIP2 case, no sonic anemometers co-located with the profiling lidars were available. Moreover, all the WINDCUBE lidars at WFIP2 operated in profiling mode using slant beams rather than in a purely vertical stare mode. Therefore, modeling the spectra of the line-of-sight velocity measured by these instruments is not trivial, as most of the spectral models are valid for
either the purely horizontal or vertical components of wind speed, and projecting these models can lead to variance contamination (Newman et al., 2016). As a consequence, we further extend this method and we estimate the optimal sample length $N$ to





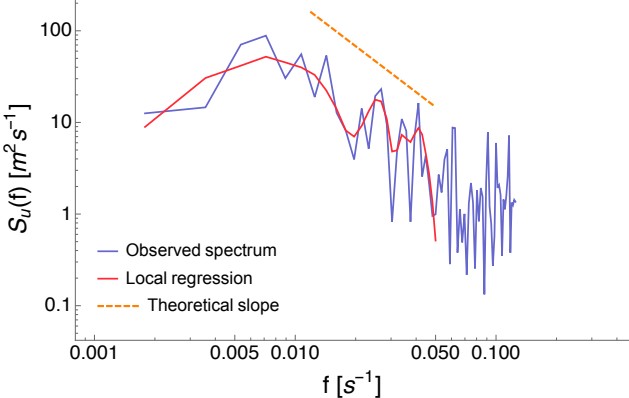

**Figure 2.** Example of local regression of an experimental spectrum of the line-of-sight velocity measured by one of the four beams of the WINDCUBE v1 lidar at Wasco Airport at WFIP2. The dashed line shows the theoretical $-5/3$ slope of the spectrum in the inertial subrange.

use in the retrieval of $\epsilon$ by determining the extension of the inertial subrange as the maximum in the curve representing a local regression of the spectrum of the line-of-sight velocity measured by the lidars. In doing so, we do not need to know the precise functional form for the spectrum of the measured radial velocity in an arbitrary slant direction. Using the dataset described in Bodini et al. (2018), with sonic anemometers co-located with lidars, we tested different local regression techniques, and we

select the robust LOESS technique (Cleveland, 1979), with a span of 15% of the total number of data points in each spectrum, which provided the best agreement ($R^2 > 0.95$) with the $\epsilon$ values obtained from the fine-tuning with the estimates from the sonic anemometers. In the determination of the maximum of the local regression curve, we leave out frequencies greater than 0.05 Hz, which are most affected by instrumental noise (Frehlich, 2001). An example of local regression of an experimental lidar spectrum at WFIP2 is shown in Figure 2.

Finally, the contribution due to instrumental noise needs to be considered. The observed variance $\sigma_v^2$ in (5) can be thought as a combination of three different contributions, which can be considered as independent of one other (Doviak et al., 1993):

$$\sigma_v^2 = \sigma_w^2 + \sigma_e^2 + \sigma_d^2 \tag{7}$$

where $\sigma_w^2$ is the contribution from atmospheric turbulence at the scales the lidar can measure (Brugger et al., 2016), $\sigma_e^2$ is due to the instrumental noise, and $\sigma_d^2$ is related to the variation in the aerosol terminal fall velocity within the sampled volume,

which can safely be ignored since the particle fall speed is typically very low ($< 1\,\mathrm{cm\,s^{-1}}$). The contribution of instrumental noise $\sigma_e^2$ can be written as a function of the signal-to-noise ratio (SNR) Pearson et al. (2009):

$$\sigma_e^2 = \frac{\Delta\nu^2\sqrt{8}}{\alpha N_p}\left(1 + \frac{\alpha}{\sqrt{2\pi}}\right)^2 \tag{8}$$

where $\Delta\nu$ is the signal spectral width; $\alpha$ is the ratio of the lidar photon count to the speckle count (Rye, 1979), which can be calculated as a function of the bandwidth $B$ as $\alpha = \frac{SNR}{\sqrt{2\pi}}\frac{B}{\Delta\nu}$. The accumulated photon count $N_p$ can be calculated as

$N_p = SNRnM$, with $n$ the number of lidar pulses which are averaged to get a profile, and $M$ the number of points sampled



within a single range gate. Therefore, $\epsilon$ can be determined as

$$\epsilon = 2\pi \left(\frac{2}{3a}\right)^{3/2} \left(\frac{\sigma_v^2 - \sigma_e^2}{L_N^{2/3} - L_1^{2/3}}\right)^{3/2} \tag{9}$$

with the accurate choice of the appropriate sample length $N$, as described.

## 2.4 Turbulence dissipation rate from scanning Doppler lidar

Turbulence dissipation rate from the scanning Doppler lidars is estimated using the azimuth structure function method (Frehlich et al., 2006; Krishnamurthy et al., 2011). The structure function from the radial velocity estimates can be used to estimate turbulence dissipation rate, the integral length scale and the velocity variance, assuming a theoretical model for isotropic wind fields. In our approach, corrections for turbulence measurements have been considered to address the complications due to the inherent volumetric averaging of radial velocity over each range gate, the noise of the lidar data, and the assumptions required
to estimate the effects of smaller scales of motion on turbulence quantities. Both the scanning lidars have an azimuth scan rate of $1°\mathrm{s}^{-1}$, and an accumulation time of $1\,\mathrm{s}$, which determine an azimuth spacing $\Delta\Phi = 1°$.

  The structure function $\hat{D}_{wgt}$ of the mean Doppler lidar velocity perturbations, $\hat{v}'$, in the azimuth direction is given by

$$\hat{D}_{wgt}(R, kR\Delta\Phi, \theta) = \frac{1}{N_s - k} \sum_{j=1}^{N_s - k} [\hat{v}'(R, (j-1)\Delta\Phi, \theta) - \hat{v}'(R, (j+k-1)\Delta\Phi, \theta)]^2 - 2\sigma_e^2(R), \tag{10}$$

where $\Delta\Phi$ is the azimuth angular spacing between adjacent Doppler velocity estimates, and $N_s$ is the number of velocity
measurements for the sector scan. The estimation error is uncorrelated with the pulse-weighted velocity because each estimate is produced with different lidar pulses (assuming no multi-scattering effects); therefore, the velocity error variance $\sigma_e^2(R)$ is only a function of the range gate (Krishnamurthy, 2008).

  For homogeneous von Kármán turbulence over a two-dimensional plane, the following model (Hinze, 1959; Frehlich et al., 2006) for the structure function is valid:

$$D_v(r, s) = 2\sigma_v^2[\Lambda(\frac{p}{L_o}) + \Lambda_D(\frac{p}{L_o})(1 - \frac{r^2}{p^2})], \tag{11}$$

where $r$ denotes the distance along a fixed laser beam, $s = R(\phi_1 - \phi_2)$ is the transverse coordinate, $p = (r^2 + s^2)^{1/2}$, $L_o$ is the outer scale of turbulence, which is proportional to the integral length scale $L_i$, $\Lambda(x)$ is the universal function and

$$\Lambda_D(x) = \frac{x^{4/3}}{2^{1/3}\Gamma(1/3)} K_{2/3}(x) = 0.3x^{2/3} K_{2/3}(x). \tag{12}$$

  Assuming that the averaged radial velocity can be written as a function of the instantaneous radial velocity and an effective
spatial filter in terms of the pulse-weighting function and range-gate length of the lidar (Frehlich et al., 2006), the Doppler lidar azimuth structure function can be modeled as

$$D_{wgt}(s, \sigma, L_o) = 2\sigma^2 G_a(\frac{s}{\Delta p}, \mu, \zeta), \tag{13}$$





where $s = R(\phi_1 - \phi_2)$, $\sigma$ is the standard deviation of the transverse velocity fluctuations and $G_a(\eta, \mu, \zeta)$ is the derived model based on weighted velocity estimates and the von Kármán model, as provided in Equation (46) of Frehlich et al. (2006) and fully derived in Krishnamurthy (2008).

The parameters $\sigma$ and $L_o$ are estimated by minimizing the error between the lidar derived structure function $\hat{D}_{wgt}(R, kR\Delta\Phi, \theta)$ and the model estimates $\hat{D}_{wgt}(s, \sigma, L_o)$. The dissipation rate can then be estimated by (Hinze, 1959)

$$\epsilon = (0.933)\frac{\sigma^3}{L_o}. \tag{14}$$

Although the assumption of homogeneous isotropic turbulence is not valid for every condition, the effect of anisotropy on the azimuth structure function is small (Krishnamurthy et al., 2011). Therefore, with an accurate choice of the scan angle and vertical resolution, the isotropic assumption can be relaxed in this algorithm for complex terrain applications. Using the selected scans described in the previous section, we retrieve $\epsilon$ from the WINDCUBE 200S and the Halo Streamline lidars every 15 minutes.

## 3 Results and Discussion

Turbulence dissipation rate has been retrieved from the five sonic anemometers at the Physics Site, four profiling lidars and two scanning lidars. This extensive network of measurements at WFIP2 allows for a unique assessment of the spatial and temporal variability, at various scales, of $\epsilon$ in complex terrain.

### 3.1 Microscale variability of turbulence dissipation rate in complex terrain

The analysis of the retrievals of turbulence dissipation rate from the five 10-m sonic anemometers, all located within a $\sim 4\,\mathrm{km}^2$ area at the Physics Site (Figure 1, panel c), allows insight into the microscale variability of $\epsilon$ in the surface layer in complex terrain.

To gain first insights on the evolution of $\epsilon$ within the Physics Site, a portion of the time series of $\epsilon$ and correspondent wind speed from the five sonic anemometers can be analyzed (Figure 3). Turbulence dissipation rate exhibits variability of at least 3 orders of magnitude over a diurnal cycle, with higher values generally observed during daytime conditions, and lower values at night. However, the magnitudes observed in the diurnal cycle of $\epsilon$ show a considerable variability among different days, with the minimum values during the night of the calendar day 176, when high winds were recorded, being similar to the maximum magnitudes observed during daytime convective conditions on day 178, when the wind was more quiescent. Moreover, although the five considered towers are all located within a $\sim 4\,\mathrm{km}^2$ area, $\epsilon$ still shows a considerable variability among the different sonic anemometers. This variability is particularly accentuated at night (especially for the night between the calendar days 178 and 179), when $\epsilon$ varies of more than an order of magnitude within the considered microscale region.

Given this distinct variability of $\epsilon$ at different times of the day, the impact of atmospheric stability conditions can be additionally investigated throughout the $\sim 13$ months of measurements at the Physics Site. To understand whether a systematic difference in the microscale variability of $\epsilon$ during different atmospheric stability conditions can be found, we calculated, at





**Figure 3.** Time series from 24 June 2016 00 UTC to 27 June 2016 00 UTC comparing $\epsilon$ (panel a) and wind speed (panel b) from five sonic anemometers at 10 m AGL at the Physics Site. Data have been smoothed with a 30-min running mean. Blue shaded areas show local nighttime conditions, while orange areas show local daytime periods.

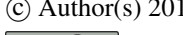



**Table 3.** Standard deviation of the distribution of the ratios between $\epsilon_i$ from each sonic anemometer and the average $\bar{\epsilon}$ from all the five sonic anemometers, for different atmospheric stability conditions.

| Met tower | std($\epsilon_i/\bar{\epsilon}$) stable conditions | std($\epsilon_i/\bar{\epsilon}$) unstable conditions |
|---|---|---|
| P03 | 0.94 | 0.84 |
| P04 | 0.78 | 0.66 |
| P05 | 0.74 | 0.69 |
| P09 | 0.95 | 0.89 |
| P10 | 0.75 | 0.64 |
| Mean | 0.83 | 0.74 |

each time, the ratio between $\epsilon$ from each sonic anemometer and the average $\epsilon$ (at that time) from all the five sonic anemometers. We then classified these ratios based on atmospheric stability, quantified as the median value of the Obukhov length from the five sonic anemometers. For each sonic anemometer, we estimate the variability of $\epsilon$ in different stability conditions in terms of the standard deviation of the distribution of these $\epsilon$ ratios, as reported in Table 3. For all five sonic anemometers, the standard

deviation is higher during stable conditions compared to unstable conditions, with mean (across the five anemometers) values of 0.83 and 0.74, respectively. On average, in the surface layer, at the small spatial scales sampled within the Physics Site, $\epsilon$ shows a 12% larger variability during nighttime stable conditions compared to daytime convective conditions.

Along with atmospheric stability, topographic features can have an impact on the variability of turbulence dissipation rate, and the high-density array of meteorological towers at the Physics Site represents and ideal candidate to explore this relation

at the microscale. Figure 4 shows the wind rose obtained from the 10-m sonic anemometer on the P03 meteorological tower at the Physics Site (the wind roses from the four other sonic anemometers are qualitatively similar to the one shown here, and are reported in the Supplement). The prevailing wind directions at the Physics Site follow the dominant west-east direction of the Columbia River Gorge. As the wind at the site is almost always slightly south-westerly, it is interesting to study whether differences in turbulence dissipation rate can be found as the wind flows from the western to the eastern sides of the Physics

Site. The five meteorological towers at the Physics Site can be divided into the sub-group on the western side of the Physics Site (towers P03 and P09) and the remaining ones east of this cluster, which we will refer to as "eastern" (towers P04, P05, and P10). We note that the far eastern side of the Physics Site includes an 80-m tower (Wilczak et al., 2019). We can first assess the topographic impact on the microscale variability of $\epsilon$ in terms of the distribution of the ratio between the mean $\epsilon$ from the groups of sonic anemometers on the two sides of the Physics Site (Figure 5). A systematic bias is observed in the values of $\epsilon$

on the two sides of the Physics Site, with the median value of turbulence dissipation on the eastern side being only 73% of the median $\epsilon$ on the western side. These differences may be due to the drainage flows and channeling frequently observed at night at this site. Topography reveals to have an impact on the variability of turbulence dissipation rate even at small spatial scales, of the order of $2\,\mathrm{km}$ in this case.





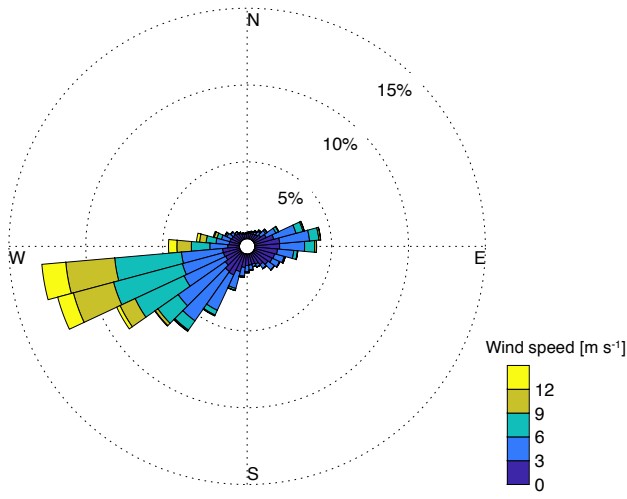

**Figure 4.** Wind rose computed from the data recorded by the 10-m sonic anemometer on the meteorological tower P03 at the Physics Site, from 29 March 2016 to 15 May 2017.

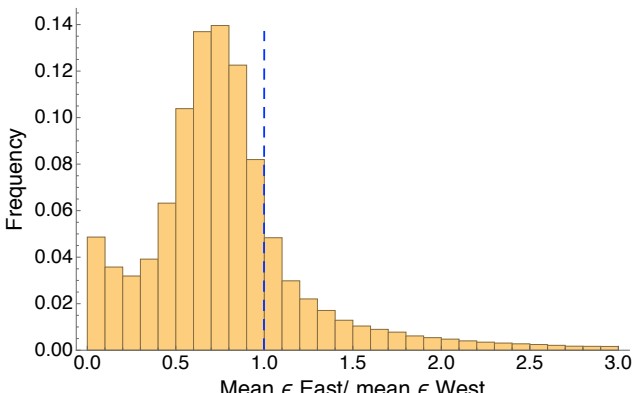

**Figure 5.** Histogram of the ratio of the average $\epsilon$ retrieved from the three 10-m sonic anemometers on the eastern side of the Physics Site (towers P04, P05, and P10) to the average of $\epsilon$ retrieved from the two sonic anemometers on the western side of the Physics Site (towers P03, P09). The vertical dashed line shows the 1.0 ratio, which would indicate no difference, on average, in $\epsilon$ between the two sides of the site.

To confirm this result, the correlation between $\epsilon$ retrievals from all the possible pairs of meteorological towers at the Physics Site can be studied (Figure 6, panel a). Stations which are close by (separation $< 1\,\text{km}$) and on the same side of the Physics Site show high correlation coefficients ($R > 0.75$). When considering pairs of stations on opposite sides of the Physics Site (with separations between 1 and $2\,\text{km}$), we find smaller correlations ($R < 0.7$) for turbulence dissipation rate, as reasonable since the spatial separation between the towers increases. However, when looking at the correlation between the retrievals from the two sonic anemometers on the western side of the Physics Site, which have the highest separation ($\sim 2.2\,\text{km}$), we



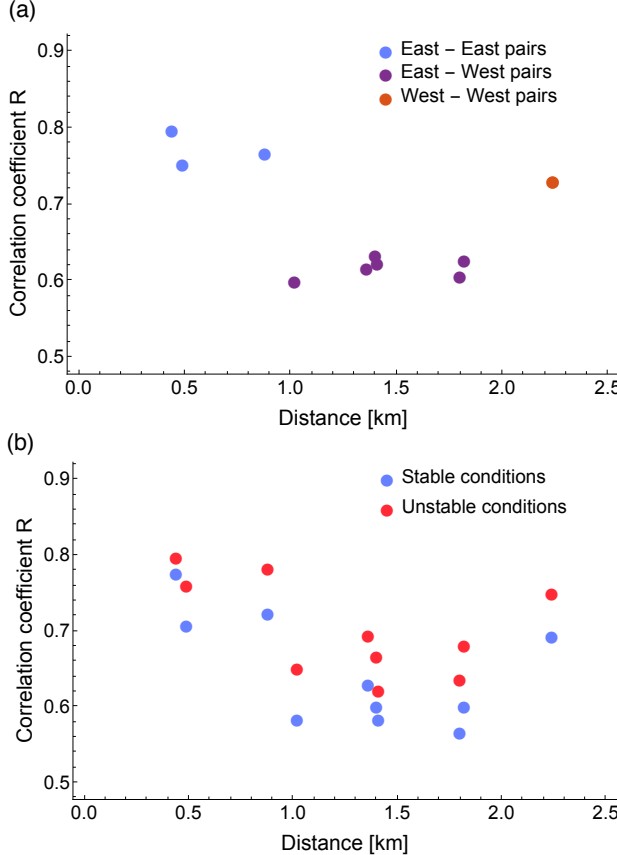

**Figure 6.** Correlation coefficient $R$ between $\log(\epsilon)$ from different pairs of 10-m sonic anemometers at the Physics Site as a function of the separation between the single meteorological towers. In panel a, different colors are used for pairs of towers both on the western side of the Physics Site, both on the eastern side, or on both sides. In panel b, data points are classified as a function of atmospheric stability.

still find a relatively high correlation coefficient ($R > 0.7$). Larger separations do not represent the only dominant factor in determining a progressive reduction of the coefficient of correlation, as the specific interaction between the atmospheric flow and the topographic features in complex terrain seem to be capable of modifying the spatial evolution of correlation between turbulence dissipation at different locations.

5   The relationship between correlation coefficient and separation can also provide a confirmation of the larger variability of $\epsilon$ observed during stable conditions. When calculating the correlation coefficient between $\epsilon$ values classified in stable and unstable conditions, calculated in terms of the median value of the Obukhov length (Figure 6, panel b), we find systematically larger values of $R$ during unstable conditions compared to stable conditions, at every spatial separation. During quiescent stable conditions, the increased variability of $\epsilon$ even at the microscale determines a reduced correlation throughout the site. On the
10  other hand, when considering the evolution of the correlation coefficient as a function of the elevation difference among the meteorological towers, no systematic trend can be found (plot shown in the Supplement).





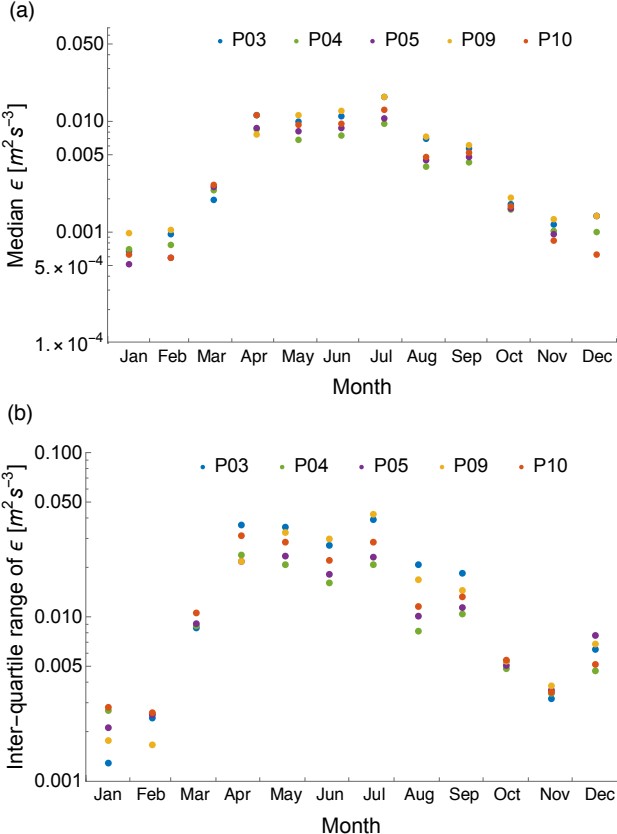

**Figure 7.** Median $\epsilon$ value for each calendar month and each considered sonic anemometer (panel a), and correspondent inter-quartile range (panel b).

Finally, the temporal variability of turbulence dissipation rate at the microscale can be assessed in terms of the annual cycle of $\epsilon$. The increased daytime convection combined with stronger, on average, winds during the summer cause larger turbulent mixing, which in turns leads to higher values of dissipation rates compared to winter months. Figure 7 (panel a) quantifies this process by showing how the median value of $\epsilon$ varies as a function of the month of the year, for each of the five stations at the
5   Physics Site. The annual cycle of wind speed is shown in the Supplement. $\epsilon$ shows a clear annual cycle in the surface layer, with median $\epsilon$ values over an order of magnitude larger in summer than winter, at all the five locations considered within the Physics Site. As a consequence, the inter-quartile range of $\epsilon$ also reveals an annual cycle (Figure 7, panel b), with a larger range of variability in summer than winter, again with differences of orders of magnitude.



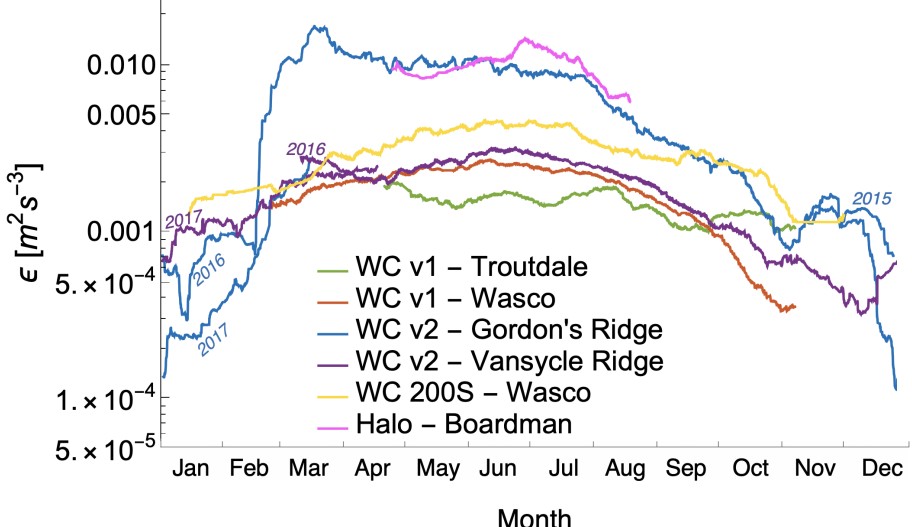

**Figure 8.** Low-pass filtered (with a 15-day moving average) time series of $\epsilon$ from the four considered profiling lidars and the two scanning lidars as a function of the calendar day, at $100\,\mathrm{m}$ AGL.

## 3.2 Mesoscale variability of turbulence dissipation rate in complex terrain

While the analysis of the heavily-instrumented Physics Site provides a unique long-term dataset to explore the microscale variability of turbulence dissipation rate in the surface layer, the four wind profiling lidars and the two scanning lidars allow for an evaluation of the variability of $\epsilon$, at higher altitudes relevant for wind energy, in a region spanning $\sim 300\,\mathrm{km}$.

5 The annual cycle in turbulence dissipation rate found at the Physics Site can also be detected from the retrievals, at higher altitude, from the lidars at the mesoscale. Figure 8 shows the time series of $\epsilon$ from the different lidars, with a low-pass filter (15-day moving window) applied to filter out the high-frequency and diurnal fluctuations and focus on the seasonal trend. For the lidars at Gordon's Ridge and at Vansycle Ridge, which were deployed for more than a year, two time series are plotted for the overlapping calendar days from different years. The time series of the seasonal cycle of wind speed for the

10 different lidars is included in the Supplement. The time series confirm that turbulence dissipation shows a distinct seasonal variability: $\epsilon$ is, on average, much higher during the summer, when strong convection increases turbulence production and, consequently, dissipation. Average $\epsilon$ values during winter are about one order of magnitude lower than what is observed in summer. Measurement records longer than a single year would be beneficial to filter out possible variations of $\epsilon$ linked with specific weather conditions, which, together with snow melting on the ground, possibly impacted the abrupt increase in average

15 $\epsilon$ values at Gordon's Ridge in the spring.

 Moreover, the smoothed time series also reveals how turbulence dissipation rate at Boardman and Gordon's Ridge is, except for winter months, much larger than at the other locations, with the average time series at the other locations showing, on average, almost one order of magnitude lower values of $\epsilon$. To explore why $\epsilon$ shows much larger values at these locations,

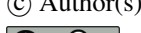


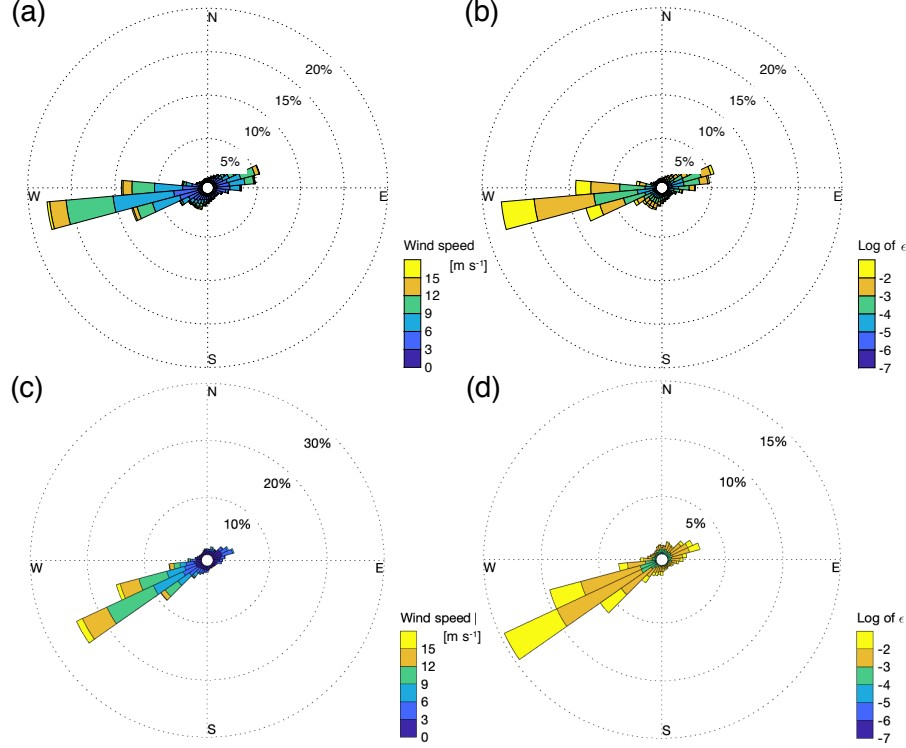

**Figure 9.** Wind roses at $100\,\mathrm{m}$ AGL from the WINDCUBE v2 at Gordon's Ridge (panel a) and the Halo Streamline at Boardman (panel c), and correspondent turbulence dissipation roses at the same altitude (panels b and d).

Figure 9 shows the wind roses and the correspondent roses of turbulence dissipation rate, at $100\,\mathrm{m}$ AGL, for the WINDCUBE v2 and the Halo Streamline lidar. At Gordon's Ridge, westerly winds are the prevailing pattern, with some north-easterly winds being the second most common situation. The highest values for $\epsilon$ are measured during westerly wind conditions, while cases with easterly winds rarely have $\epsilon > 10^{-3}\mathrm{m^2s^{-3}}$. When the wind flows from the west, the location of the WINDCUBE v2

5    lidar is at the downwind edge of a higher-altitude mountain area ($\sim 6\,\mathrm{km}$ wide) with particularly complex topography. The dissipation of eddies in the wake of an obstacle is larger, leading to higher values of $\epsilon$.

    With the dominant southwesterly wind, the lidar at Boardman turns out to be located downwind (about $15\,\mathrm{km}$) of a large wind farm. Wind farm wakes are associated with reduced wind speed and increased turbulence, which can have important impacts on wind energy production downwind (S. Lissaman, 1979; Nygaard, 2014). Wind speed deficits from wind farm

10    wakes have been observed using SAR (Christiansen and Hasager, 2005; Hasager et al., 2006), radars (Hirth et al., 2015) and aircraft measurements (Platis et al., 2018) up to $25\,\mathrm{km}$ downwind of the plants. Systematic turbulence measurements that far downwind of wind farms have not yet been made. However, turbulence dissipation measurements $2-3$ rotor diameters in the wake of a single turbine (Lundquist and Bariteau, 2015) showed an elevated level of $\epsilon$. Therefore, the increased dissipation





aloft observed at Boardman is likely due to the increased turbulence aloft in the wind farm wake. Wind roses and turbulence dissipation roses for the other lidars are included in the Supplement.

The seasonal variability of turbulence dissipation can be additionally investigated by considering the differences in the average daily conditions of $\epsilon$ throughout the year. Figures 10 and 11 show the average diurnal climatology of turbulence
dissipation rate at the various locations of the four profiling lidars and the two scanning lidars, respectively. The left column shows the average climatology for the summer, calculated as average conditions from 1 June to 31 August. For the profiling lidars at Gordon's Ridge and Vansycle Ridge, and the scanning lidar at Wasco, which were also deployed during winter months, the panels on the right column show the average daily cycle for the winter, using $\epsilon$ retrievals from 1 December to the end of February. For all the lidars, we neglect the heights where less than $15\%$ of data within the considered season are available (the
complete data availability is shown in the Supplement). In all the locations, turbulence dissipation rate shows a clear diurnal cycle, with higher values during daytime convective conditions, and lower values at night, with differences greater than one order of magnitude, especially in summer. The inter-comparison between the plots from the different lidars also confirms the impact of topography in determining much higher average values of $\epsilon$ at Gordon's Ridge compared to what is recorded at the other locations. In particular, daytime summer values are about one order of magnitude higher than what is found from the
other lidars. At Boardman, large average values of dissipation are found aloft at night. In fact, the increased turbulence in the wind farm wake can be further advected during nighttime stable periods, when stronger stratification is found in the boundary layer. The comparison between the summer climatologies (left panels) with the winter ones (right panels) reveals how larger values of $\epsilon$ are found during the summer compared to what is found in the wintertime diurnal climatology, when daytime $\epsilon$ values are about two to three orders of magnitude lower, and with a much weaker difference between daytime and nighttime
average conditions. It is reasonable to expect that the increased diurnal convection during the summer months determines much stronger turbulent mixing, which in turns causes higher values of turbulence dissipation.

## 4   Conclusions

Although turbulence is a fundamental transport mechanism in the atmospheric boundary layer, current numerical weather prediction models are limited in their representation of turbulence, where a local equilibrium between production and dissipation
($\epsilon$) of turbulence is assumed. The error introduced by the parametrization of $\epsilon$ has been shown to be responsible for up to 50% of the variance of hub-height wind speed predicted by models. The detailed study of observations in the surface layer has a great potential for reducing the uncertainty in our understanding of turbulence dissipation rate. Although methods to retrieve $\epsilon$, at least from in situ measurements, have been known for decades, comprehensive analysis of the spatial and temporal variability of $\epsilon$ using data from instruments covering wide regions had not been fully explored to date. In this study we have presented
an extensive assessment of the variability, both in space and time, of turbulence dissipation rate in complex terrain at both the microscale and mesoscale, using measurements from both in situ and remote sensing instruments. The impact of topography and other forcings, like large wind farms, on the variability of $\epsilon$ has been captured at the different sampled scales.





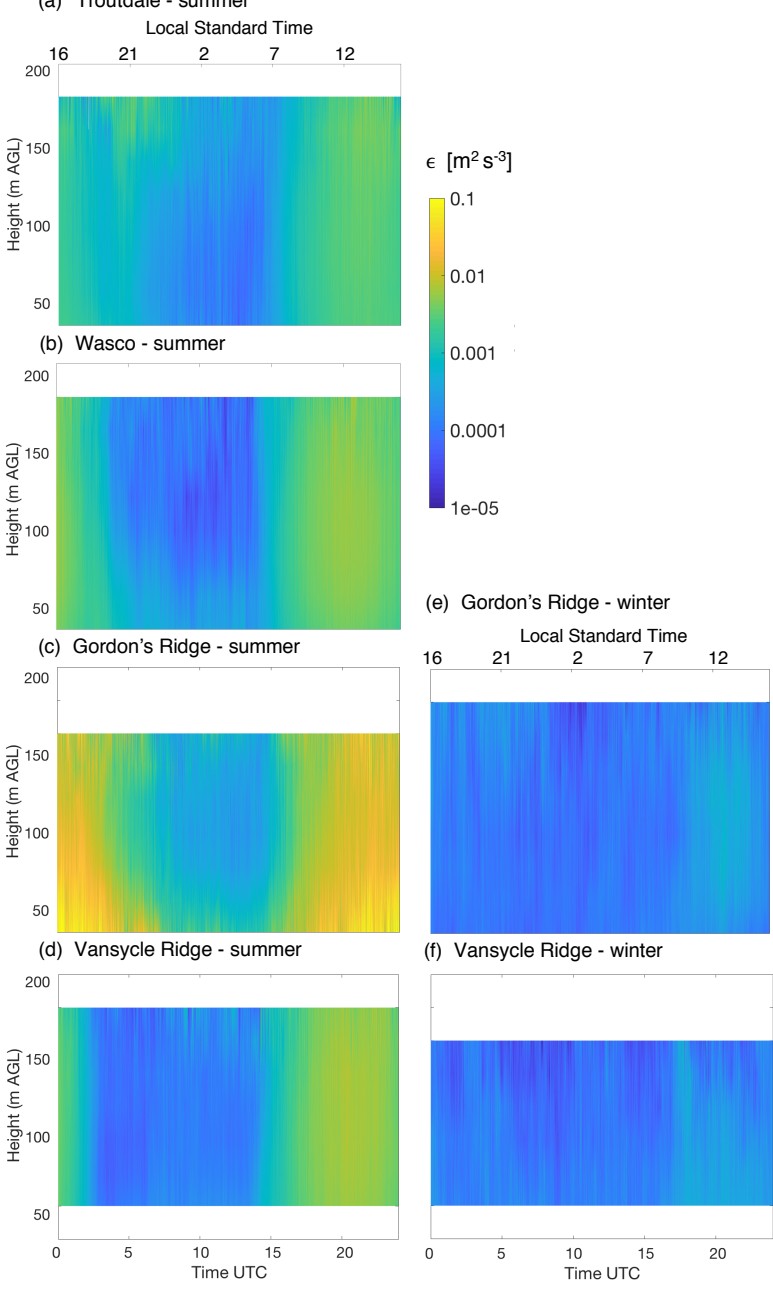

**Figure 10.** Average diurnal climatology of $\epsilon$ for the summer (1 June - 31 August) from the WINDCUBE v1 at Troutdale (panel a), the WINDCUBE v1 at the Wasco Airport (panel b), the WINDCUBE v2 at Gordon's Ridge (panel c), and the WINDCUBE v2 at Vansycle Ridge (panel d). Average diurnal climatology for the winter (1 December - 28/29 February) from the WINDCUBE v2 at Gordon's Ridge (panel e), and the WINDCUBE v2 at Vansycle Ridge (panel f). At this site, LST = UTC - 8.



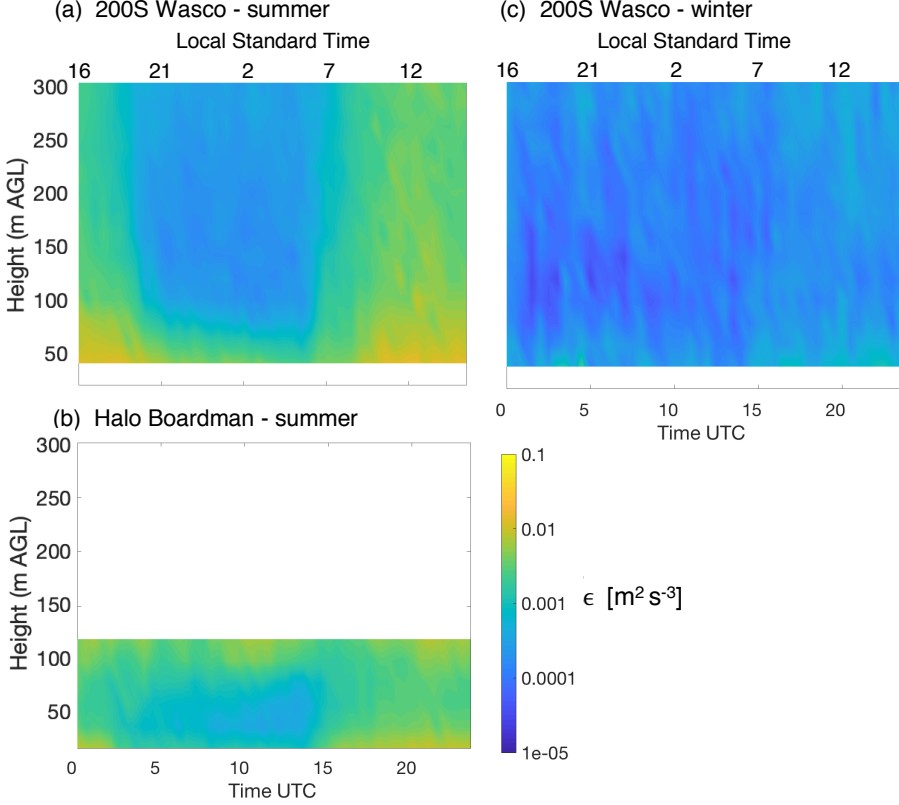

**Figure 11.** Average diurnal climatology of $\epsilon$ for the summer (1 June - 31 August) from the WINDCUBE 200S at Wasco (panel a), the Halo Streamline at Boardman (panel b). Average diurnal climatology for the winter (1 December - 28/29 February) from the WINDCUBE 200S at Wasco (panel c). At this site, LST = UTC - 8.

Turbulence dissipation rate has been calculated from five 10-m sonic anemometers, four wind profiling lidars and two scanning lidars deployed at the WFIP2 field campaign in the Columbia River Gorge and Basin from Fall 2015 to Spring 2017. The sonic anemometers were all located in an area with an extension of approximately $2\,km$ x $2\,km$, and they therefore allow for an assessment of the variability of $\epsilon$ in the surface layer at the microscale. More homogeneous turbulence across the

5 investigated region is caused by the convective mixing during the day. On the other hand, considerable differences (up to one order of magnitude) in $\epsilon$ are found at night when comparing retrievals of $\epsilon$ from the different meteorological towers. On average, $\epsilon$ is 12% more variable during nighttime stable conditions than during unstable convective conditions. Systematic differences emerged from $\epsilon$ measured on the western and eastern sides of the Physics Site, in correspondence with the dominant westerly wind pattern, thus suggesting the possible impact of topography in triggering the variability of $\epsilon$. The change of correlation

10 between $\epsilon$ in different locations is not fully determined purely by spatial separation, as topographic features maintain an importance in influencing it. Therefore, the representation of turbulence dissipation rate in complex terrain, especially during nighttime stable conditions, needs to be extremely localized to fully capture the turbulence variability in the surface layer.





The variability of $\epsilon$ at the mesoscale can be analyzed from the 100-m altitude retrievals from the four wind profiling lidars and the two scanning lidars, which were deployed over a region $\sim 300\,\mathrm{km}$ wide. For the profiling lidars, the retrieval approach proposed in Bodini et al. (2018) has here been further refined and tested to derive $\epsilon$ without the need of in situ measurements co-located with the lidars. The profiling lidar located at the topographically complex Gordon's Ridge site systematically detected

$\epsilon$ values which, on average, were over one order of magnitude higher than what was measured by the profiling lidar deployed in the gentler Troutdale, Wasco Airport and Vansycle Ridge sites. The dominant westerly winds at the site resulted in the location of this lidar to be on the downwind edge of an orographic complex, therefore experiencing a strong increase in turbulence production, and consequently dissipation. Similarly, the scanning lidar located at Boardman showed higher values of $\epsilon$ due to the increased turbulence in the wake of a wind farm.

The extensive duration of the WFIP2 field campaign has allowed for the evaluation of the annual cycle of $\epsilon$: the increased convective mixing in summer determines higher values of $\epsilon$ compared to the typically more quiescent winter conditions, with an average difference which can reach one order of magnitude, both at the microscale and at the mesoscale, in the surface layer and above. We have determined the impact of this seasonal cycle on the average diurnal climatology of $\epsilon$. Overall, $\epsilon$ is, on average, up to three orders of magnitude higher in summer compared to winter. The diurnal cycle, with higher values of $\epsilon$

during daytime convective conditions and lower values at night is much stronger during the summer, where diurnal differences in $\epsilon$ values are of about two orders of magnitude, while the reduced daytime convection during wintertime leads to a more uniform average daily climatology, with less than one order of magnitude of difference between daytime and nighttime values of $\epsilon$.

Future work can explore and compare the variability of $\epsilon$ from other datasets in different topographic conditions, as well as in

the offshore environment (Peña et al., 2009; Canadillas et al., 2010; Türk and Emeis, 2010). Once this systematic assessment of the variability of turbulence dissipation has been completed from different regions, all the insights on the spatial and temporal variability of $\epsilon$ should be incorporated into numerical weather prediction model representations of turbulence. As this variability appears to be dependent on several different atmospheric and topographic factors, machine learning techniques, which have already been successfully applied to a broad range of complex atmospheric problems (Sharma et al., 2011; Xingjian et al.,

2015; Alemany et al., 2018; Gentine et al., 2018), could provide a reliable representation of $\epsilon$.

*Data availability.* The data of the sonic anemometers and wind Doppler lidars at the WFIP2 field campaign are publicly available at https://a2e.energy.gov/data.

*Author contributions.* JKL, LKB and MP helped designing and carrying out the field measurements. NB analyzed the data from the sonic anemometers and the profiling lidars and made the figures, in close consultation with JKL. RK analyzed the data from the scanning lidars.

NB wrote the paper, with significant contributions from JKL and RK. All the coauthors contributed to the refining the paper text.





*Competing interests.* The authors declare that they have no conflict of interest.

*Acknowledgements.* The authors thank Josh Aikins, Joseph Lee, Clara St. Martin, Jessica Tomaszewski, and Rochelle Worsnop for helping with the lidar deployment at WFIP2. The authors thank Dr. David Cook for deploying some of the 10-m meteorological towers whose data have been used in this work. The authors also thank Dr. Chris Hocut from Army Research Laboratory and Dr. Harindra J Fernando from

5    University of Notre Dame for providing the scanning Lidar data at Boardman site. The authors also thank Sonia Wharton at the Lawrence Livermore National Laboratory for providing data from the lidar at Vansycle Ridge. LLNL funding came from the A2e Mesoscale Physics and Inflow: WFIP2 (project # 01.03.1.302), U.S. DOE Office of Energy Efficiency and Renewable Energy Wind Energy Technologies. The authors appreciate Mr. Matthieu Boquet's and Dr. Ludovic Thobois' efforts to provide some of the technical specifications of the WINDCUBE v1 and v2 used in our analysis. JKL and NB are supported by the National Science Foundation (AGS-1554055) under the

10   CAREER program. RK is supported by internal funds from the University of Notre Dame for contributions to this paper. LKB and MP were supported by the Department of Energy, Office of Energy Efficiency and Renewable Energy Wind Power Program. The Pacific Northwest National Laboratory is operated for DOE by the Battelle Memorial Institute under Contract DE-A06-76RLO1830. This work was authored (in part) by NREL, operated by the Alliance for Sustainable Energy, LLC, for the U.S. DOE, under Contract No. DE-AC36-08GO28308, with funding provided by the U.S. DOE Office of Energy Efficiency and Renewable Energy Wind Energy Technologies.



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
