# Peer review of "Spatial and temporal variability of turbulence dissipation rate in complex terrain"

_Atmospheric Chemistry and Physics, 2018_

## Referee Comment (RC1) · Anonymous Referee #1 · 6 Jan 2019

General Comments: This paper presents a unique set of observations of turbulence dissipation rates, both on the mesoscale across the Columbia River Gorge, and on a scale about the size of a model grid cell. The variability between sites is shown to exist in time and in space. Overall, this is an important study, to understand the differences that can be expected on different spatial scales. The paper is very well written, with figures and supplemental material to support the discussion.

Specific Comments: There is a lot of mention of topography being the reason for differences and variability between sites, but what is it about the sites' topography? Is it, for example, complexity of topography in a ∼1km radius? Or maybe the slopes of the sites? In order to attribute the variability to topography, we need to know more about that topography. Furthermore, with no explanation of how the topography impacts dis-

sipation, it is hard to negate the possibility that different instruments produce different magnitudes of dissipation and its diurnal cycle.

In the conclusion, discuss what the microscale variability means for mesoscale modeling. How do models need to account for this subgridscale variability?

P3 1st full paragraph: Mention that Troutdale is on W side of Cascades, with the other sites in the Columbia Basin to the east of the Range P3L14: What height are the towers? P3 2nd full paragraph: The sentence about the wind farms/turbine in the larger region is out of place in the paragraph about the sonics at the physics site. Move this sentence to the paragraph before, or break into pieces in previous paragraph and this one. P4: Mention that Wasco, Gordon's Ridge and Vansycle Ridge are on the east side of the Cascades, in the Basin. How far apart from each other are the sites? P5L11: why does a fast scan rate need to be removed? Figure 2: identify the maximum of the local regression, which is used for N Figure 3: Can you show an addition day on either end, to show the more-typical diurnal cycles? P12: How does the topography impact the east vs west side of the Physics Site? What is the variability in terrain (there's 50m between the highest and lowest points, but is there a ridge, is it a uniform slope, which direction is higher/lower, etc)? P12: Is this analysis done only when winds are from the southwest? If not, it would be interesting to see if the easterly winds contribute to the ratios greater than 1 in Fig 5. P16L6: mention in the text the height shown in the figure P18L2: There are wind turbines east of Wasco; do these directions show elevated dissipation rates under easterly winds? It's hard to see in the supplement figure. P18L14: what is it about Gordon's ridge that makes its topography special, compared to the other sites?

Supplement Fig 3: add interquartile range, like Fig 7 Supplement 4: Why is Troutdale SO small in summer?

Technical Corrections: P2L28: observational assessments P3L19: Physics P5L3: should one of the 260m top heights be different? Why specify Vansycle Ridge to 260m

AGL? P6L16: "to the their" P12L9: an ideal candidate

---

## Referee Comment (RC2) · Anonymous Referee #2 · 8 Jan 2019

The paper analyses wind and turbulence measurement data obtained from in-situ and remote sensing instrumentation during the WFIP2 campaign in the Columbia River Gorge in the North-West of the USA. It aims at describing the dissipation rate of the turbulence kinetic energy of the flow in orographically complex terrain at various spatial and temporal scales and under different thermal stratification. The authors refer to the need of a better description and parametrisation of turbulence, esp. turbulence dissipation rate, in numerical weather forecast models of different resolutions, and hope that their analysis can provide some insight into the characteristics of turbulence in complex terrain and help the modellers to parameterise it better in their codes.

The paper is well written and good to understand for readers familiar with the subject. For example, when reading the manuscript, several times I had a question which was

soon getting answered later in the text! Very comfortable! Although their qualitative findings (e.g. the patchiness/intermittency of turbulence) are neither new nor surprising, the authors provide a thorough and helpful quantitative analysis. I recommend publication of the paper after the authors have commented on my few points / questions in order make the paper even more comprehensive and complete:

P. 7: the importance of the choice of the sampling size N is correctly emphasised. Could you mention typical (and extreme) example sizes for LN in your data set ? So, what are the dimensions of the turbulence inertial subrange ? The end is given in Fig.2 but where does it typically start ? At f = 0.01 Hz, as Fig.2 possibly suggests?

P.12, Table 3: Why don't you include the neutral flow conditions ? And how frequent do the three stratification classes occur ? In other words, how large is the sampling size for your statistics ?

P.12 bottom and P.13, Fig. 5 and P.14, line 3, and P. 20 lines 7-9 : It did not get clear to me which differences in topography between the "west" and the "east" parts of the "Physics Site" may cause the biased distribution in the mean dissipation rates shown in the figure. Could you provide some more details here ? Or maybe there are other causes for that ? The sample sizes should be large enough to not account for that (then arbitrary) bias, shouldn't it?

P.13, Fig. 5: In my view even more striking than the bias is the difference in the tails of the distribution of the mean dissipation rates displayed in the figure: in about 1% of the cases $<\varepsilon \text{east}>$ is between 2.5 and 3.0 times larger than $<\varepsilon \text{west}>$; whereas $<\varepsilon \text{west}>$ is at least 10 times larger than $<\varepsilon \text{east}>$ in 5% of the cases (or at least 5 times larger in 8.5% of the cases). So the tails are in line with the bias: there are more frequent and stronger turbulence "outbreaks" in the western domain compared to the eastern domain. Why ? What causes this strong difference in intermittency ? Is there a topographic feature which could create some coherency (structure) in the turbulence in the western domain which is absent in the eastern part for the prevailing westerly

winds ?

---

## Referee Comment (RC3) · Anonymous Referee #3 · 29 Jan 2019

The paper under review by Bodini et al., reports turbulent kinetic energy (TKE) dissipation rate measurement in the Columbia river gorge using sonic anemometers, scanning Doppler lidars and profiling Doppler lidars.

Page 7. Line 1-5: It is not clear how TKE dissipation rate could be estimated from "line-of-sight" velocity. Please provide detailed clarification.

Most important point: All the methods used in the paper use some sort of coarse graining or filtering over the actual fluctuating velocity signal. Given that TKE dissipation rate is after all a small scale quantity, the authors could have tried to directly estimate TKE dissipation rate = $2 \nu * \langle s_{ij} s_{ij} \rangle$ where $s_{ij}$ is the fluctuating strain rate tensor. Or with a constant temperature anemometer they could have measured the surrogate TKE dissipation rate $2 \nu * \langle (du/dt)^2 \rangle$.

Finally, intermittent behavior of TKE dissipation rate is well known. Despite the large database this work creates the paper here is rather observational and does not report causality of the observations, or connect the scale dependence of TKE dissipation rate to reasonably well established turbulence theories on TKE dissipation rate (see Turbulence by U. Frisch). This is a weakness of the paper and needs to be addressed.

---

## Author Comment (AC1) · 26 Feb 2019

*In this document, the reviewer's comments are in black, the authors' responses are in red.*

The authors thank the reviewer for their comments.

The paper under review by Bodini et al., reports turbulent kinetic energy (TKE) dissipation rate measurement in the Columbia river gorge using sonic anemometers, scanning Doppler lidars and profiling Doppler lidars.

Page 7. Line 1-5: It is not clear how TKE dissipation rate could be estimated from "line-of-sight" velocity. Please provide detailed clarification.
We have added the specification that the variance of the line-of-sight velocity measured by the lidars is "averaged across the different beams". Since the method used is not new and it is not the main focus of the present study, we think the interested reader can find a complete and detailed explanation of the method in O'Connor et al. 2010 and Bodini et al. 2018, as stated in the paragraph.

Most important point: All the methods used in the paper use some sort of coarse graining or filtering over the actual fluctuating velocity signal. Given that TKE dissipation rate is after all a small scale quantity, the authors could have tried to directly estimate TKE dissipation rate = 2 \nu * <sij sij> where sij is the fluctuating strain rate tensor. Or with a constant temperature anemometer they could have measured the surrogate TKE dissipation rate 2 \nu * < (du/dt)^2 >.
We appreciate your suggestion. However, for the sonic anemometer data, TKE dissipation rate has been derived using either structure functions or energy spectra in a long tradition of studies (Champagne et al. 1977, Oncley et al. 1996, Piper et al. 2004, Muñoz-Esparza et al. 2018, among others), and good agreement has been found with super high-frequency measurements from hot-wire anemometers (Piper et al. 2004).
   Moreover, for the lidars, calculating the strain rate tensor has at least two inherent problems (see reference list below):
  1) the range-gate averaged measurement should be within the inertial sub-range of turbulence;
  2) the u, v, w measurements must be instantaneous in space and time.
For short-pulsed lidars like the WINDCUBE used in this study, the range-gate is small enough to show that it usually lies within the inertial sub-range (Kumer et al. 2016). However, getting the instantaneous 3D components is extremely difficult in a complex flow field, unless one uses synchronized Doppler Lidars, which was not the case for our experiment.
   For our current lidar dataset, we therefore decided to use the methods we explained in the respective sections (structure-function and velocity variance method), which have previously been shown to compare well with in-situ TKE dissipation rate estimates. The assumption of locally isotropic flow is assumed to not have a high impact on the average dissipation rate estimates over significant temporal averages (10 minutes).

References:
- Liu S, Meneveau C, Katz J (1994) On the properties of similarity subgrid-scale models as deduced from measurements in a turbulent jet. J Fluid Mech 275: 83–119
- Meneveau C, Katz J (2000) Scale invariance and turbulence models for large-eddy simulation. Annu Rev Fluid Mech 32: 1–32

- Sheng J, Meng H, Fox RO (2000) A large eddy PIV method for turbulence dissipation rate estimation. Chem Eng Sci 55: 4423–4434
- Sharp KV, Adrian RJ (2001) PIV study of small-scale flow structure around a Rushton turbine. AIChE J 47(4): 766–778
- Krishnamurthy, R., Calhoun, R., & Fernando, H. (2010). Large-Eddy simulation-based retrieval of dissipation from coherent Doppler Lidar data. Boundary-layer meteorology, 136(1), 45-57

Finally, intermittent behavior of TKE dissipation rate is well known. Despite the large database this work creates the paper here is rather observational and does not report causality of the observations, or connect the scale dependence of TKE dissipation rate to reasonably well established turbulence theories on TKE dissipation rate (see Turbulence by U. Frisch). This is a weakness of the paper and needs to be addressed.

We agree that our study has an observational nature, as we state in the Introduction of the paper. To expand our discussion of causality, as the reviewer requests, we have described in more detail why we think that topography has an impact on the variability of dissipation at the microscale, in terms of different slopes of the terrain, and also provided a detailed description (as well as an additional map in the Supplement) of the topography at Gordon Ridge, which we think is responsible for the large values of dissipation measured at that site. We believe that the explanation of the causality of what we see in our observations has now been improved.

We have also added a reference to Turbulence by Frisch and A First Course in Turbulence by Tennekes and Lumley in Sections 2.2 and 3.2 to provide additional theoretical references to our methods and results. We have also made an explicit reference to the intermittency of turbulence and the theories to describe it in Section 3.1, in the description of the variability of $\varepsilon$ in the time series at the Physics Site: "This variability can be connected to the intermittent nature of turbulence dissipation rate, for which a multifractal theory has been developed (Frish 1995)."

---

## Author Comment (AC2) · 26 Feb 2019

**In this document, the reviewer's comments are in black, the authors' responses are in red.**

**The authors thank the reviewer for their thoughtful and productive comments.**

The paper analyses wind and turbulence measurement data obtained from in-situ and remote sensing instrumentation during the WFIP2 campaign in the Columbia River Gorge in the North-West of the USA. It aims at describing the dissipation rate of the turbulence kinetic energy of the flow in orographically complex terrain at various spatial and temporal scales and under different thermal stratification. The authors refer to the need of a better description and parametrization of turbulence, esp. turbulence dissipation rate, in numerical weather forecast models of different resolutions, and hope that their analysis can provide some insight into the characteristics of turbulence in complex terrain and help the modellers to parameterise it better in their codes.

The paper is well written and good to understand for readers familiar with the subject. For example, when reading the manuscript, several times I had a question which was soon getting answered later in the text! Very comfortable! Although their qualitative findings (e.g. the patchiness/intermittency of turbulence) are neither new nor surprising, the authors provide a thorough and helpful quantitative analysis. I recommend publication of the paper after the authors have commented on my few points / questions in order make the paper even more comprehensive and complete:

**Thank you for finding our paper interesting and easy to read!**

P. 7: the importance of the choice of the sampling size N is correctly emphasized. Could you mention typical (and extreme) example sizes for LN in your data set? So, what are the dimensions of the turbulence inertial subrange? The end is given in Fig.2 but where does it typically start? At f = 0.01 Hz, as Fig.2 possibly suggests?

We have added the following sentence to the description of the method: "The distribution of sample size values we obtain is between 20s (5th percentile) and 300s (95th percentile)." In Figure 2, we have also added a vertical line to the figure to identify the maximum of the local regression, and we have changed the caption of the figure accordingly.

P.12, Table 3: Why don't you include the neutral flow conditions? And how frequent do the three stratification classes occur? In other words, how large is the sampling size for your statistics? Neutral conditions occurred less than 7% of the time. As such, we do not think they represent a sample large enough to introduce a separate category. We have now specified the % of cases which showed neutral stratification in Section 2.1.

P.12 bottom and P.13, Fig. 5 and P.14, line 3, and P. 20 lines 7-9 : It did not get clear to me which differences in topography between the "west" and the "east" parts of the "Physics Site" may cause the biased distribution in the mean dissipation rates shown in the figure. Could you provide some more details here ? Or maybe there are other causes for that? The sample sizes should be large enough to not account for that (then arbitrary) bias, shouldn't it?

Thank you for this comment – please see our answer to the next comment.

P.13, Fig. 5: In my view even more striking than the bias is the difference in the tails of the distribution of the mean dissipation rates displayed in the figure: in about 1% of the cases <reast>

is between 2.5 and 3.0 times larger than <ewest>; whereas <ewest> is at least 10 times larger than <eeast> in 5% of the cases (or at least 5 times larger in 8.5% of the cases). So the tails are in line with the bias: there are more frequent and stronger turbulence "outbreaks" in the western domain compared to the eastern domain. Why? What causes this strong difference in intermittency? Is there a topographic feature which could create some coherency (structure) in the turbulence in the western domain which is absent in the eastern part for the prevailing westerly winds?

Yes, we agree that the data suggests that the local topography is a prime cause of this intermittency. Thank you for pointing out that we should have described the topography of the Physics Site in more detail as follows:

- We have added the following sentences to the description of the division of the five meteorological towers in two sub-groups in Section 3.1: "An analysis of the topography of the region reveals two distinct sets of terrain characteristics. The terrain to the west of the sub-group of towers on the western side of the Physics Site (towers P03 and P09) has slopes that reach 60%, and the average slopes larger than 6%. On the contrary, the remaining towers east of this cluster, which we will refer to as "eastern" (towers P04, P05, and P10), are surrounded by a terrain with more gentle slopes, which are on average less than 6% and never exceed 25%."
- We have also made this point more explicit later in the Section: "The presence of steep topography increases the variability of turbulence dissipation rate even at small spatial scales".
- We have also modified a sentence in the Conclusions as follows: "Systematic differences emerged in ε measured on the western and eastern sides of the Physics Site, the former being located downwind of terrain with larger slopes compared to the latter, thus suggesting the possible impact of terrain slope in triggering the variability of ε."
- Finally, we have added 10-m elevation contour lines to the detailed map of the Physics Site in Figure 1:

---

## Author Comment (AC3) · 26 Feb 2019

In this document, the reviewer comments are in black, the authors responses are in red.

The authors thank the reviewer for their detailed review and useful suggestions to improve the quality of our work.

**General Comments:**

This paper presents a unique set of observations of turbulence dissipation rates, both on the mesoscale across the Columbia River Gorge, and on a scale about the size of a model grid cell. The variability between sites is shown to exist in time and in space. Overall, this is an important study, to understand the differences that can be expected on different spatial scales. The paper is very well written, with figures and supplemental material to support the discussion.

**Thank you for finding our results interesting!**

**Specific Comments:**

There is a lot of mention of topography being the reason for differences and variability between sites, but what is it about the sites' topography? Is it, for example, complexity of topography in a  $\sim$ 1km radius? Or maybe the slopes of the sites? In order to attribute the variability to topography, we need to know more about that topography. Furthermore, with no explanation of how the topography impacts dissipation, it is hard to negate the possibility that different instruments produce different magnitudes of dissipation and its diurnal cycle.

Thank you for pointing out that we should have described the topography of the Physics Site in more detail as follows:

- We have added the following sentences to the description of the division of the five meteorological towers in two sub-groups in Section 3.1: "An analysis of the topography of the region reveals two distinct sets of terrain characteristics. The terrain to the west of the sub-group of towers on the western side of the Physics Site (towers P03 and P09) has slopes that reach 60%, and the average slopes larger than 6%. On the contrary, the remaining towers east of this cluster, which we will refer to as "eastern" (towers P04, P05, and P10), are surrounded by a terrain with more gentle slopes, which are on average less than 6% and never exceed 25%."
- We have also made this point more explicit later in the Section: "The presence of steep topography increases the variability of turbulence dissipation rate even at small spatial scales".
- We have also modified a sentence in the Conclusions as follows: "Systematic differences emerged in ε measured on the western and eastern sides of the Physics Site, the former being located downwind of terrain with larger slopes compared to the latter, thus suggesting the possible impact of terrain slope in triggering the variability of ε."
- Finally, we have added 10-m elevation contour lines to the detailed map of the Physics Site in Figure 1:

We have also improved the description of the topography of Gordon Ridge, to give a more detailed explanation for the larger dissipation values recorded at the site, and we have added a detailed map of the area in the Supplement (see specific comment below).

In the conclusion, discuss what the microscale variability means for mesoscale modeling. How do models need to account for this subgridscale variability?

We agree that this is an essential question, but the answer requires extensive research, and a detailed response is beyond the scope of the work presented in this manuscript. To motivate additional research on the topic, we have rephrased some sentences of our conclusions as follows: "Assessing the spatial and temporal variability of  $\varepsilon$  within a typical grid cell of a mesoscale model will provide further insights into the validity of sub-grid scale  $\varepsilon$  parameterization schemes during various atmospheric stability conditions. As this variability appears to be dependent on several different atmospheric and topographic factors, complex techniques are likely needed to provide accurate spatial representations of  $\varepsilon$  over a mesoscale grid. Sophisticated tools such as physics-driven machine learning techniques (Sharma et al. 2011, Xingjian et al. 2015, Alemany et al. 2018, Gentine et al. 2018) are paving the path to capture the microscale variability of  $\varepsilon$  in mesoscale models accurately."

P3 1st full paragraph: Mention that Troutdale is on W side of Cascades, with the other sites in the Columbia Basin to the east of the Range

We have added the specification "... Troutdale (the only site on the western side of the Cascades)" in the paragraph.

P3L14: What height are the towers?

The tower heights range from 10 to 80m. However, since in this study we are only using the 10-m sonic anemometers (as specified a few lines later), we think it is better to omit this detail to not confuse the reader, and to just reference the overall WFIP2 observational paper (Wilczak et al. 2019) for those interested in details about the overall experiment.

P3 2nd full paragraph: The sentence about the wind farms/turbine in the larger region is out of place in the paragraph about the sonics at the physics site. Move this sentence to the paragraph before, or break into pieces in previous paragraph and this one.

We have moved to the previous paragraph the following sentence: "Extensive arrays of wind turbines are located on the northern side of the Columbia River and on the south-western part of the studied region.". We have added to this paragraph this sentence: "Several wind turbines are located east of the Physics Site."

P4: Mention that Wasco, Gordon's Ridge and Vansycle Ridge are on the east side of the Cascades, in the Basin. How far apart from each other are the sites?

We have added specifications such as "in an area within the Columbia Basin" and "on the eastern side of the Cascades" for these locations. Moreover, the maps in Figure 1 show these locations and a scale bar, so that the interested reader can determine the distance between the different sites.

P5L11: why does a fast scan rate need to be removed?

To calculate accurate turbulence statistics from the azimuth structure-function method, the effective sensing volume transverse to the lidar beam needs to be much smaller than the range-gate size (Frehlich et al., 2006). During WFIP2, the fast scan rates violated this assumption and hence these scans were ignored in our analysis to provide accurate estimates. In the manuscript, we have added a reference to Frehlich et al., 2006 in the sentence for the reader who might be interested in more details about the method.

Figure 2: identify the maximum of the local regression, which is used for N We have added a vertical line to the figure to identify the maximum of the local regression. We have also changed the caption of the figure accordingly.

Figure 3: Can you show an addition day on either end, to show the more-typical diurnal cycles? We find that each particular day has some sort of unique behavior (see plot below), dependent on the complexity of the various quantities that impact turbulence dissipation rate. Even adding additional days would not really define a typical diurnal cycle, which is instead shown in Figures 10 and 11 for all the lidar locations.